

# Patterns of genetic structuring at the northern limits of the Australian smelt (*Retropinna semoni*) cryptic species complex

Md Rakeb-Ul Islam[1,*], Daniel J. Schmidt[1,*], David A. Crook[2,*] and Jane M. Hughes[1,*]

[1] Australian Rivers Institute, Griffith University, Brisbane, Australia
[2] Research Institute for Environment and Livelihoods, Charles Darwin University, Darwin, NT, Australia
[*] These authors contributed equally to this work.

## ABSTRACT

Freshwater fishes often exhibit high genetic population structure due to the prevalence of dispersal barriers (e.g., waterfalls) whereas population structure in diadromous fishes tends to be weaker and driven by natal homing behaviour and/or isolation by distance. The Australian smelt (Retropinnidae: *Retropinna semoni*) is a native fish with a broad distribution spanning inland and coastal drainages of south-eastern Australia. Previous studies have demonstrated variability in population genetic structure and movement behaviour (potamodromy, facultative diadromy, estuarine residence) across the southern part of its geographic range. Some of this variability may be explained by the existence of multiple cryptic species. Here, we examined genetic structure of populations towards the northern extent of the species' distribution, using ten microsatellite loci and sequences of the mitochondrial cyt *b* gene. We tested the hypothesis that genetic connectivity among rivers should be low due to a lack of dispersal via the marine environment, but high within rivers due to dispersal. We investigated populations corresponding with two putative cryptic species, SEQ-North (SEQ-N), and SEQ-South (SEQ-S) lineages occurring in south east Queensland drainages. These two groups formed monophyletic clades in the mtDNA gene tree and among river phylogeographic structure was also evident within each clade. In agreement with our hypothesis, highly significant overall $F_{ST}$ values suggested that both groups exhibit very low dispersal among rivers (SEQ-S $F_{ST}$ = 0.13; SEQ-N $F_{ST}$ = 0.27). Microsatellite data indicated that connectivity among sites within rivers was also limited, suggesting dispersal may not homogenise populations at the within-river scale. Northern groups in the Australian smelt cryptic species complex exhibit comparatively higher among-river population structure and smaller geographic ranges than southern groups. These properties make northern Australian smelt populations potentially susceptible to future conservation threats, and we define eight genetically distinct management units along south east Queensland to guide future conservation management. The present findings at least can assist managers to plan for effective conservation and management of different fish species along coastal drainages of south east Queensland, Australia.

Corresponding author
Md Rakeb-Ul Islam,
r.islam@griffithuni.edu.au

## INTRODUCTION

Genetic structure in aquatic fauna is strongly influenced by the characteristics of the ambient environment. Freshwater species typically exhibit higher levels of genetic differentiation than those living in estuarine or marine habitats (*Ward, Woodwark & Skibinski, 1994*; *Sharma & Hughes, 2009*). These greater levels of genetic structure in freshwater fish from different drainages are the result of the isolating nature of drainage systems and relatively smaller population sizes compared with marine species (*McGlashan & Hughes, 2002*; *Huey, Baker & Hughes, 2010*). Strong to moderate genetic structure is observed in many freshwater fish species suggesting restricted gene flow and limited dispersal among populations (*Leclerc et al., 2008*; *Pereira, Foresti & Oliveira, 2009*; *Huey, Baker & Hughes, 2010*). Movement by obligate freshwater organisms is limited to the water column and the freshwater environment, preventing inter-catchment dispersal via the sea (*Burridge et al., 2008*; *Hughes, Schmidt & Finn, 2009*; *Bernays et al., 2015*). In addition, various in-stream barriers to dispersal including waterfalls, dam walls, habitat heterogeneity, dried river reaches, and steep catchment gradients may act to restrict gene flow among populations within catchments (*Amoros & Bornette, 2002*; *Huey, Baker & Hughes, 2010*). As a consequence of the physical limitations to dispersal in freshwater environments, populations of aquatic organisms are often highly differentiated both among and within catchments (*McGlashan & Hughes, 2000*; *Hughes, 2007*; *Sharma & Hughes, 2009*). For a particular species, it is essential to understand the levels of population differentiation, genetic diversity, and rates of gene flow among populations for proper conservation and management of freshwater ecosystems (*Geist, 2011*).

The southern smelt (Retropinnidae: *Retropinna spp.*) is a common fish species distributed throughout the rivers of south-eastern Australia (*McDowall, 1996*). Individuals reach a maximum length of about 100 mm total length (TL), although adults are usually 50–60 mm TL (*Pusey, Kennard & Arthington, 2004*). Australian smelts are currently formally recognised as two described species *R. semoni* Weber, and *R. tasmanica* McCulloch, but recent genetic analyses have identified a complex of five or more cryptic species across their geographic range based on allozymes, microsatellites and mitochondrial DNA data (*Hammer et al., 2007*; *Hughes et al., 2014*; *Schmidt, Islam & Hughes, 2016*). Otolith chemistry studies in the southern part of their distribution have shown that Australian smelt exhibit a range of life history patterns, including freshwater residency, facultative diadromy and estuarine residency (*Crook, Macdonald & Raadik, 2008*; *Hughes et al., 2014*). In inland regions of Australia, large numbers of Australian smelt have been observed moving upstream through fishways (e.g., *Baumgartner & Harris, 2007*) and the species is widely described as potamodromous (i.e., migration within freshwater for the purpose of breeding) (e.g., *Rolls, 2011*). Nonetheless, *Woods et al. (2010)* found strong genetic structure among inland populations of Australian smelt and suggested low levels of dispersal in at

least some regions. These differences between studies could reflect differing life-histories among the cryptic species.

In most studies to date, diadromous behaviour has been shown to facilitate genetic connectivity among river catchments and typically results in "isolation-by-distance" (IBD) patterns of population genetic structure (*Keenan, 1994*; *Jerry & Baverstock, 1998*). In Australian smelt, however, there is strong genetic differentiation among catchments across the southern part of the range—even among populations containing diadromous individuals, suggesting high retention of fish within estuaries and a lack of marine dispersal (*Hughes et al., 2014*). The aim of the current study was to examine patterns of genetic connectivity of populations in the north of the geographic range of Australian smelt, which contains two putative cryptic species ('SEQ', 'CEQ' *sensu*. *Hammer et al., 2007*) that differ from those studied in detail previously ('MTV', 'SEC' *sensu*. This study focuses on the SEQ lineage which was further subdivided into northern and southern groups (SEQ-N, SEQ-S) following *Page & Hughes (2014)*. Mitochondrial DNA sequence data combined with nuclear data from 10 microsatellite loci were used to test the hypotheses that, (i) northern *R. semoni* would display high population structure among rivers similar to southern populations; and (ii) that genetic structure within rivers would be low due to dispersal within rivers.

## MATERIALS AND METHODS

### Sampling strategy

A total of 389 individual samples were collected from 15 locations in south-east Queensland, Australia (Fig. 1; Table 1A). Samples were collected using a hand—held seine net from an upstream and a downstream site from each of eight river systems (except the Noosa River—downstream only). We aimed to collect at least 30 individuals per site but this was not always possible, as the species was not abundant in all rivers. Fin clips or entire individuals were placed in 95% ethanol in the field and stored prior to preparation for analysis. All procedures were carried out according to Australian Ethics Commission protocol number ENV/23/14/AEC.

### Molecular methods

Genomic DNA was extracted from fin tissue using the DNeasy Blood and Tissue kits (Qiagen, Germantown, MD, USA) following the manufacturer's directions. Microsatellite markers developed for *R. semoni* were amplified and genotyped using primers developed by *Islam, Schmidt & Hughes (2017)*. Ten loci were screened across all individuals. The ten loci were BS18, BS3, BS4, BS5, BS20, BS21, BS22, BS24, BS8 and MS24. Microsatellite screening was carried out in 10 μl polymerase chain reaction (PCR) consisting of 0.5 μl of genomic DNA, 0.2 mM reverse primer, 0.05 mM tailed forward primer, 0.2 mM tailed fluorescent tag (either FAM, VIC, NED or PET, Applied Biosystems), 1 × PCR buffer (Astral Scientific, Caringbah, Australia) and 0.02 units of *taq* polymerase (Astral Scientific, Caringbah, Australia). The following basic thermocycler settings for PCR were performed: initial denaturation at 94 °C for 4 min, followed by 35 cycles at 94 °C for 1 min, 57 °C for 30 s, 72 °C for 1 min and a final extension at 72 °C for 7 min. Fluorescently
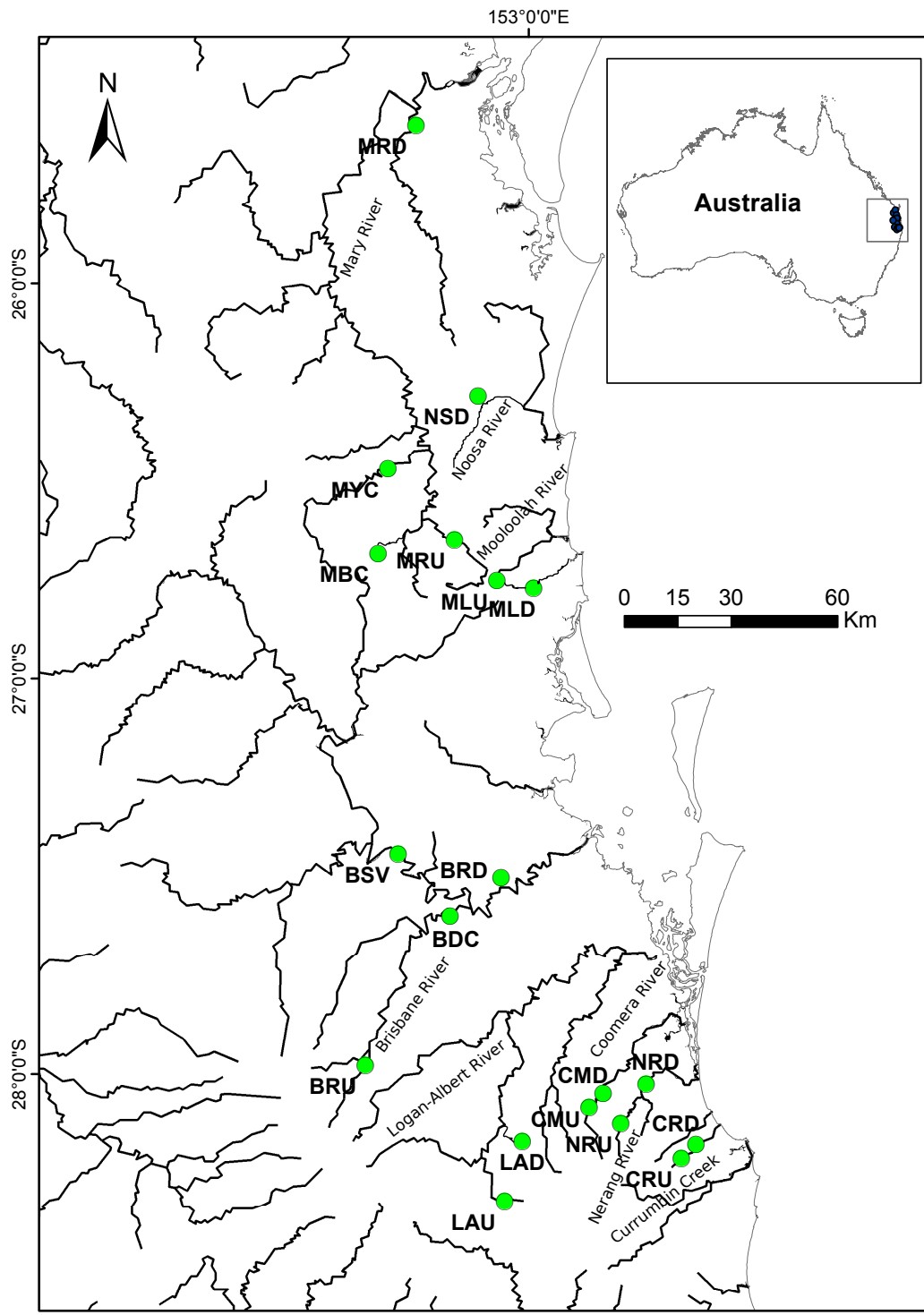

**Figure 1 Localities where specimens of *R. semoni* were collected during this study in south-east Queensland, Australia.** The green circle denotes the sampling site. See Tables 1A and 1B for site codes and locations.

**Table 1** **(A) Summary of sample information, genetic diversity indices and population specific $F_{ST}$ for microsatellite loci of Australian smelt. (B) Summary of sample information for mitochondrial DNA sequences of Australian smelt.** Number of samples used for genetic analysis ($N$), mean number of alleles per population ($N_A$), observed heterozygosity ($H_O$), expected heterozygosity ($H_E$), mean inbreeding index ($F_{IS}$).

**(A)**

| Group name | Sampling site | Site code | Longitude (E) | Latitude (S) | $N$ | $N_A$ | $H_O$ | $H_E$ | $F_{IS}$ | Population specific $F_{ST}$ |
|---|---|---|---|---|---|---|---|---|---|---|
| SEQ-N | Tinana | MRD | 152°42′57.8″ | 25°36′04.3″ | 26 | 8.90 | 0.686 | 0.733 | 0.066 | 0.196 |
| SEQ-N | Mary_upper | MRU | 152°48′47.9″ | 26°38′55.5″ | 29 | 10.90 | 0.779 | 0.826 | 0.059 | 0.136 |
| SEQ-N | Noosa_lower | NSD | 152°52′21.4″ | 26°17′05.7″ | 30 | 6.10 | 0.576 | 0.617 | 0.067 | 0.340 |
| SEQ-N | Mooloolah_lower | MLD | 153°0′44.64″ | 26°46′18.83″ | 16 | 4.60 | 0.670 | 0.604 | −0.115 | 0.390 |
| SEQ-N | Mooloolah_upper | MLU | 152°55′13.1″ | 26°45′07.9″ | 32 | 6.40 | 0.609 | 0.566 | −0.079 | 0.334 |
| SEQ-S | Brisbane_lower | BRD | 152°55′49.9″ | 27°30′16.05″ | 32 | 9.50 | 0.754 | 0.755 | 0.002 | 0.191 |
| SEQ-S | Brisbane_upper | BRU | 152°35′13.5″ | 27°58′43.9″ | 32 | 8.80 | 0.760 | 0.763 | 0.005 | 0.216 |
| SEQ-S | Logan-Albert_lower | LAD | 152°59′01.8″ | 28°10′15.6″ | 8 | 7.60 | 0.701 | 0.847 | 0.184 | 0.085 |
| SEQ-S | Logan-Albert_upper | LAU | 152°56′23.6″ | 28°19′19.7″ | 24 | 11.40 | 0.828 | 0.845 | 0.021 | 0.107 |
| SEQ-S | Coomera_lower | CMD | 153°11′20.9″ | 28°02′55.5″ | 24 | 13.90 | 0.839 | 0.887 | 0.054 | 0.074 |
| SEQ-S | Coomera_upper | CMU | 153°09′13.4″ | 28°05′01.8″ | 32 | 14.70 | 0.848 | 0.887 | 0.045 | 0.078 |
| SEQ-S | Nerang_lower | NRD | 153°17′52.0″ | 28°01′33.7″ | 8 | 6.40 | 0.718 | 0.798 | 0.106 | 0.156 |
| SEQ-S | Nerang_upper | NRU | 153°14′02.8″ | 28°07′29.2″ | 32 | 13.70 | 0.782 | 0.853 | 0.084 | 0.088 |
| SEQ-S | Currumbin_lower | CRD | 153°25′24.8″ | 28°10′41.9″ | 32 | 11.20 | 0.771 | 0.803 | 0.041 | 0.130 |
| SEQ-S | Currumbin_upper | CRU | 153°23′11.9″ | 28°12′49.6″ | 32 | 10.90 | 0.769 | 0.785 | 0.021 | 0.135 |

**(B)**

| Group name | Sampling site | Site code | Longitude (E) | Latitude (S) | $N$ (number of sample) |
|---|---|---|---|---|---|
| SEQ-N | Tinana | MRD | 152°42′57.8″ | 25°36′04.3″ | 2 |
| SEQ-N | Mary_upper | MRU | 152°48′47.9″ | 26°38′55.5″ | 2 |
| SEQ-N | Booloomba Creek | MBC | 152°37′10.6″ | 26°41′02.5″ | 9 |
| SEQ-N | Yabba Creek | MYC | 152°38′39.5″ | 26°28′09.3″ | 8 |
| SEQ-N | Noosa_lower | NSD | 152°52′21.4″ | 26°17′05.7″ | 2 |
| SEQ-N | Mooloolah_lower | MLD | 153°0′44.64″ | 26°46′18.83″ | 2 |
| SEQ-N | Mooloolah_upper | MLU | 152°55′13.1″ | 26°45′07.9″ | 3 |
| SEQ-S | Brisbane_lower | BRD | 152°55′49.9″ | 27°30′16.05″ | 2 |
| SEQ-S | Brisbane_upper | BRU | 152°35′13.5″ | 27°58′43.9″ | 2 |
| SEQ-S | Bundamba Creek | BDC | 152°48′04.2″ | 27°36′03.9″ | 10 |
| SEQ-S | Banks Creek | BSV | 152°40′13.2″ | 27°26′36.9″ | 10 |
| SEQ-S | Logan-Albert_lower | LAD | 152°59′01.8″ | 28°10′15.6″ | 2 |
| SEQ-S | Logan-Albert_upper | LAU | 152°56′23.6″ | 28°19′19.7″ | 2 |
| SEQ-S | Coomera_lower | CMD | 153°11′20.9″ | 28°02′55.5″ | 2 |
| SEQ-S | Coomera_ upper | CMU | 153°09′13.4″ | 28°05′01.8″ | 2 |
| SEQ-S | Nerang_lower | NRD | 153°17′52.0″ | 28°01′33.7″ | 2 |
| SEQ-S | Nerang_upper | NRU | 153°14′02.8″ | 28°07′29.2″ | 2 |
| SEQ-S | Currumbin_lower | CRD | 153°25′24.8″ | 28°10′41.9″ | 2 |
| SEQ-S | Currumbin_upper | CRU | 153°23′11.9″ | 28°12′49.6″ | 2 |

labelled PCR products were pooled and added to 10 µl of Hi-Di[TM] formamide with 0.1 µl of GeneScan[TM] 500 LIZ size standard. Fragment analysis was conducted on an ABI PRISM 3130 Genetic Analyzer (Applied Biosystems, Foster City, CA, USA) according to the manufacturer's instructions. Data were scored using GENEMAPPER version 3.1 software (Applied Biosystems, Foster City, CA, USA).

Two individuals from each of the 15 populations represented in the microsatellite study were randomly selected for mtDNA analysis except the Mooloolah_upper (MLU) site from which one additional sample was included for analysis. Samples from four additional sites not included in microsatellite analysis were also sequenced—two sites from the Mary River and two sites from the Brisbane River (Fig. 1 and Table 1B). In total 68 individuals from nineteen sites were sequenced. A 666 bp fragment of the cytochrome *b* region of the mtDNA genome was selected for sequence analysis. The primers HYPSLA and HYPSHD (*Thacker et al., 2007*) were used to amplify the region in 10 µL reaction mixtures. PCR conditions were 4 min at 95 °C, followed by 45 cycles of 30 s at 95 °C, 45 s at 53 °C, 45 s at 72 °C and a final extension cycle of 7 min at 72 °C. MtDNA sequences were edited and aligned using Geneious version 9.1.5 (*Kearse et al., 2012*).

## Data analysis
### Genetic diversity
Microsatellite genotype frequencies were checked for the presence of null alleles, large allele dropout and stuttering artefacts using MICRO-CHECKER v2.2.3 (*Van Oosterhout et al., 2004*). Tests for linkage disequilibrium (LD) and departures of genotypic proportions expected under Hardy-Weinberg Equilibrium (HWE) were undertaken with exact tests for each population and over all loci using default settings in GENEPOP v4 (*Rousset, 2008*). Probability values were corrected using standard Bonferroni correction (*Rice, 1989*) whenever multiple testing was performed.

Genetic diversity averaged across ten loci within each of the 15 population samples was calculated from observed and expected heterozygosity using ARLEQUIN v3.5.1.2 (*Excoffier & Lischer, 2010*). Inbreeding index ($F_{IS}$) was estimated in FSTAT 2.9.3 (*Goudet, 2001*).

### Population genetic structure
Genetic structure among the 15 populations was quantified through estimating pairwise and global $F_{ST}$ values in ARLEQUIN. These were tested for significant deviation from panmictic expectations by 10,000 permutations of individuals among populations. Population-specific $F_{ST}$ values were calculated using GESTE v2.0 (*Foll & Gaggiotti, 2006*) to evaluate the contribution of individual population samples to overall $F_{ST}$.

ARLEQUIN was used to evaluate the geographic structuring of genetic variation. $F_{ST}$ was calculated for each locus separately and as a weighted average over the ten microsatellite loci. Statistical significance of $F_{ST}$ was determined by 1,000 permutations of individuals among populations. Hierarchical structuring of variation was calculated using analyses of variance (AMOVA) in ARLEQUIN. Two hierarchical arrangements of the 15 populations were analysed where the highest level was either (a) two groups (species), SEQ-N (MRD, MRU, NSD, MLD, MLU) and SEQ-S (BRD, BRU, LAD, LAU, CMD, CMU, NRD, NRU, CRD, CRU) or (b) catchment division, grouped into eight rivers. These were: Mary (MRD,

MRU), Noosa (NSD), Mooloolah (MLD, MLU), Brisbane (BRD, BRU), Logan (LAD, LAU), Coomera (CMD, CMU), Nerang (NRD, NRU) and Currumbin (CMD, CMU). Three hierarchical levels of variation were analysed for each arrangement: among rivers ($F_{CT}$), among populations within rivers ($F_{SC}$) and among all populations ($F_{ST}$).

Bayesian clustering methods implemented in STRUCTURE v.2.3.1 (*Pritchard, Stephens & Donnelly, 2000*) were applied to estimate the number of genetically homogeneous clusters (*Latch et al., 2006*; *Hasselman, Ricard & Bentzen, 2013*). This programme builds genetic clusters by minimizing linkage disequilibrium and deviations from Hardy-Weinberg equilibrium expectations within clusters. All individuals were assigned to clusters without prior knowledge of their geographic origin using the admixture model with correlated allelic frequencies. Ten independent runs with the number of potential genetic clusters ($K$) from 1 to 16 were carried out to verify that the estimates of $K$ were consistent across runs. The burn-in length was set at 250,000 iterations followed by a run phase of one million iterations. The generated results were imported into the software STRUCTURE HARVESTER (*Earl & vonHoldt, 2012*) to calculate the *ad hoc* $\Delta K$ statistic (*Evanno, Regnaut & Goudet, 2005*). The $K$ value where $\Delta K$ had the highest value was identified as the most likely number of clusters.

### Analysis of isolation by distance
A test for a positive association between genetic and geographic distances (Isolation by distance (IBD)) based on microsatellite DNA loci was carried out using a Mantel test (10,000 permutations) in ARLEQUIN. Genetic distance was represented as $F_{ST}$. Stream distances were calculated between river mouths and then sample sites using Google Earth.

### Migration and gene flow
BAYESASS v1.3 was used to calculate contemporary migration rates over the past few generations, where *m* is the proportion of immigrants in a focal population *i* that arrive from a source population *j* (*Wilson & Rannala, 2003*). Migration rates were estimated for all pairs of sites and rate were reported that fell outside the 95% confidence interval simulated for uninformative data (*Wilson & Rannala, 2003*). We also used the Bayesian assignment procedure of *Rannala & Mountain (1997)*, as implemented in GENECLASS 2 (*Piry et al., 2004*) to estimate whether our samples might contain individuals that were first generation ($F_0$) immigrants from unsampled populations. Here we used the *Paetkau et al. (2004)* method to compute probabilities from 10,000 simulated genotypes to identify $F_0$ immigrants.

### Analysis of mtDNA sequence data
A neighbour-joining (NJ) tree analysis was performed using the HKY distance model in Geneious version 9.1.5 with 1,000 bootstrap replicates (*Kearse et al., 2012*). In addition to the 68 sequences generated from this study, two Genbank accessions were used, one representing *R. tasmanica*: JN232589; and one representing *R. semoni*: JN232588 (*Burridge et al., 2012*). The *R. semoni* sequence JN232588 lacks locality information (C Burridge, pers. comm., 2017), but likely belongs to a southern lineage of *R. semoni* which are known to have a closer mtDNA relationship with *R. tasmanica* than to northern lineages (*Hughes et al., 2014*).

## RESULTS

### Genetic variability and levels of differentiation

After Bonferroni correction, 3 out of 15 populations exhibited deviations from HWE in only one or two loci (Table S1). All loci were kept for further analyses since deviations were not consistent across populations. Instances of null alleles estimated using MICRO-CHECKER were rare and not consistently associated with specific loci or populations (Table S2). No evidence for genotypic linkage disequilibrium was observed between any pair of loci across all populations.

Population genetic diversity indices are shown in Table 1A. Microsatellite genetic diversity was high. Mean number of alleles per population ranged from 4.60 (MLD) to 14.70 (CMU) and heterozygosity averaged across loci ranged from 0.566(MLU) to 0.887 (CMD and CMU). Most sites exhibited positive $F_{IS}$ values, indicating that most of the populations had slight heterozygote deficits.

Most of the pairwise $F_{ST}$ values between the 15 populations were significant and ranged from $-0.018$ to $0.404$. The SEQ-N populations were more diverged from one another than the populations in the SEQ-S group. The lowest pairwise $F_{ST}$ value ($F_{ST} = -0.018; P < 0.05$) was observed between populations NRD and NRU. The highest genetic divergence ($F_{ST} = 0.404; P < 0.05$) was observed between populations NSD and MLU. Out of 105 comparisons, only two comparisons were non-significant ($P > 0.05$) and each of these pairs was from within the same river (Logan-Albert and Nerang). However, one site of each of these pairs contains a small number of samples which probably reduces the power to detect the significant $F_{ST}$ values. Generally $F_{ST}$ comparisons revealed much less divergence among populations within the same river than between populations from different rivers (Table 2).

STRUCTURE analysis revealed the highest likelihood was at $K = 8$ clusters (Average log probability of data $\text{Ln}[\text{P(DK)}] \pm \text{SD} = -15125.4 \pm 0.584618$) (Fig. 2, Table S3) indicating this as the best estimate of the true number of genetic clusters. The height of $\Delta K$ was used as an indicator of the strength of the signal detected by STRUCTURE (*Evanno, Regnaut & Goudet, 2005*). $\Delta K$ also showed the highest peak at $K = 8$, suggesting eight genetically homogeneous clusters across the sampled populations (Fig. S1) and negligible migration was observed among rivers (Fig. 2).

Strong genetic structure was observed by AMOVA. The AMOVA showed significant genetic differentiation between the two groups (SEQ-N and SEQ-S) ($F_{CT} = 0.05$), but also among populations within groups ($F_{SC} = 0.19$) (Table 3A). There were similar patterns observed in the two groups when each group (SEQ-N and SEQ-S) was analysed separately, with the $F_{CT}$ value (among rivers) higher than the $F_{SC}$ value (among sites within rivers) in both groups (Table 3Bi and 3Bii). However, the overall $F_{ST}$ values, and each of the other $F$ statistics in the hierarchy were slightly higher within the SEQ-N group than the SEQ-S group.

### Isolation-by-distance

There was a significant correlation between genetic differentiation and stream distance among populations in the SEQ-S group ($R^2 = 0.3687$, $p = 0.001$; BRD, BRU, LAD, LAU,

Islam et al. (2018), *PeerJ*, DOI 10.7717/peerj.4654

Peerj

**Table 2   Pairwise $F_{ST}$ values among all pairs of populations based on microsatellite data.** $F_{ST}$ estimates significance levels $< 0.05$ are in bold following Bonferroni correction. See Table 1A for site codes and locations.

| | MRD | MRU | NSD | MLD | MLU | BRD | BRU | LAD | LAU | CMD | CMU | NRD | NRU | CRD | CRU |
|---|---|---|---|---|---|---|---|---|---|---|---|---|---|---|---|
| MRD | 0.000 | | | | | | | | | | | | | | |
| MRU | **0.108** | 0.000 | | | | | | | | | | | | | |
| NSD | **0.358** | **0.310** | 0.000 | | | | | | | | | | | | |
| MLD | **0.250** | **0.182** | **0.362** | 0.000 | | | | | | | | | | | |
| MLU | **0.318** | **0.244** | **0.404** | **0.009** | 0.000 | | | | | | | | | | |
| BRD | **0.178** | **0.131** | **0.323** | **0.203** | **0.276** | 0.000 | | | | | | | | | |
| BRU | **0.173** | **0.136** | **0.320** | **0.218** | **0.295** | **0.013** | 0.000 | | | | | | | | |
| LAD | **0.243** | **0.101** | **0.394** | **0.248** | **0.316** | **0.157** | **0.170** | 0.000 | | | | | | | |
| LAU | **0.244** | **0.137** | **0.368** | **0.246** | **0.317** | **0.173** | **0.189** | −0.006 | 0.000 | | | | | | |
| CMD | **0.212** | **0.151** | **0.319** | **0.186** | **0.259** | **0.130** | **0.146** | **0.126** | **0.161** | 0.000 | | | | | |
| CMU | **0.203** | **0.141** | **0.293** | **0.165** | **0.236** | **0.124** | **0.139** | **0.127** | **0.164** | **0.010** | 0.000 | | | | |
| NRD | **0.261** | **0.125** | **0.393** | **0.244** | **0.316** | **0.187** | **0.191** | **0.073** | **0.082** | **0.152** | **0.141** | 0.000 | | | |
| NRU | **0.229** | **0.110** | **0.329** | **0.203** | **0.261** | **0.176** | **0.183** | **0.053** | **0.074** | **0.146** | **0.137** | −0.018 | 0.000 | | |
| CRD | **0.263** | **0.152** | **0.353** | **0.222** | **0.277** | **0.195** | **0.211** | **0.125** | **0.147** | **0.166** | **0.156** | **0.076** | **0.062** | 0.000 | |
| CRU | **0.305** | **0.178** | **0.397** | **0.270** | **0.320** | **0.248** | **0.256** | **0.120** | **0.150** | **0.215** | **0.201** | **0.073** | **0.059** | **0.011** | 0.000 |

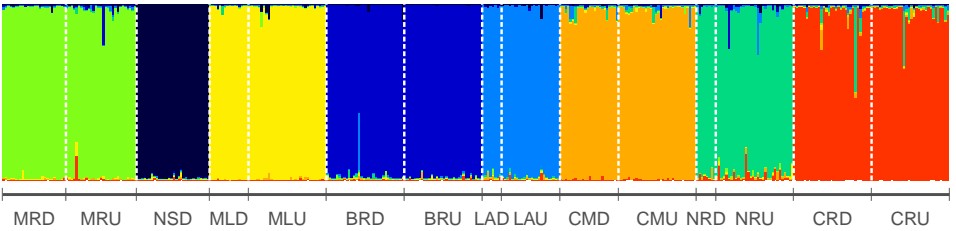

**Figure 2  Results from structure indicating individual assignment and population clustering of Australian smelt for 389 individuals from all 15 locations sampled (Table 1A) in south-east Queensland, Australia.** Individual sampling locations is listed below the figure and see Table 1A for site codes and locations. Each column represents one individual and the colours represent the probability membership coefficient of that individual for each genetic cluster. Results of Evanno's Mean LnP(D) and $\Delta K$ (Table S3; Fig. S1) indicate that the best supported $K$ values for all populations are 8.

**Table 3  Analyses of molecular variance (AMOVA) for hierarchical arrangements of the 15 sampling sites based on microsatellite data.** The hierarchical levels tested were among rivers ($F_{CT}$), among populations within rivers ($F_{SC}$) and among all populations ($F_{ST}$). The total genetic variation is shown as a percentage for each hierarchical partitioning. Hierarchical arrangement of sites: (a) sites divided into two groups: SEQ-N & SEQ-S; (b) sites divided into catchment divisions within group.

| Structure tested | | Observed partition | | F-Statistics |
|---|---|---|---|---|
| | | Variance | % of variation | |
| (a) | Based on group (SEQ-N & SEQ-S) | | | |
| | Between groups | 0.07255 Va | 5.45 | $F_{CT} = 0.05^*$ |
| | Among sites within group | 0.23313 Vb | 17.50 | $F_{SC} = 0.19^*$ |
| | Within sites | 1.02632 Vc | 77.05 | $F_{ST} = 0.23^*$ |
| (b) | Based on river | | | |
| (i) | SEQ-N group | | | |
| | Among rivers | 0.17274 Va | 19.23 | $F_{CT} = 0.19^*$ |
| | Among sites within rivers | 0.06542 Vb | 7.28 | $F_{SC} = 0.09^*$ |
| | Within sites | 0.66026 Vc | 73.49 | $F_{ST} = 0.27^*$ |
| (ii) | SEQ-S group | | | |
| | Among rivers | 0.27139 Va | 12.49 | $F_{CT} = 0.13^*$ |
| | Among sites within rivers | 0.01106 Vb | 0.51 | $F_{SC} = 0.006^*$ |
| | Within sites | 1.89107 Vc | 87.01 | $F_{ST} = 0.13^*$ |

**Notes.**
$^*P < 0.001$.

CMD, CMU, NRD, NRU, CRD, and CRU) (Fig. 3A), but not for the SEQ-N group ($R^2 = 0.0355$, $p = 0.302$; MRD, NSD, MRU, MLD and MLU) (Fig. 3B).

## Contemporary migration

Very little contemporary migration was observed among the coastal river populations. Only six sampled populations contained individuals that were identified as potential immigrants from the BAYESASS analysis. In all cases, the putative source population was the paired site within the same catchment (Table 4). In most cases, dispersal was from the upstream to the downstream site. Only individuals from Currumbin Creek were estimated to have

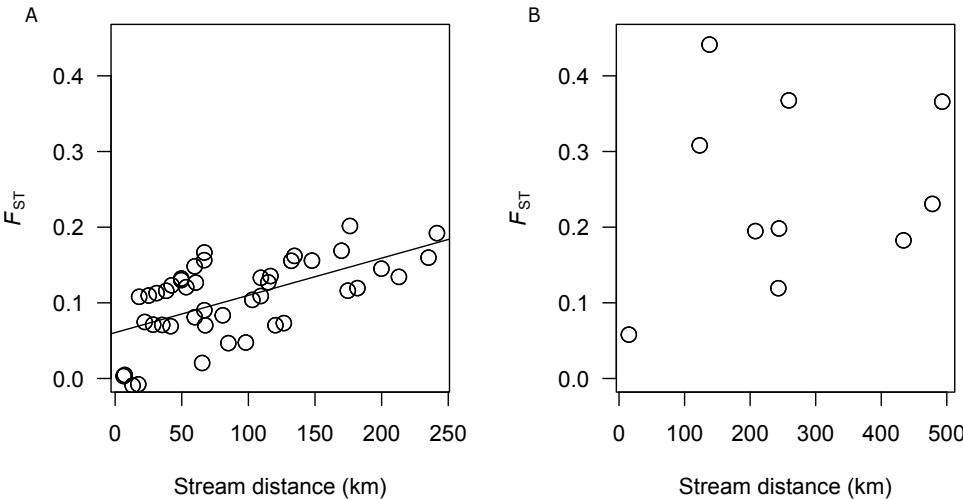

**Figure 3** (A) Analysis of isolation by distance for SEQ-S populations. (B) Analysis of isolation by distance for SEQ-N populations.

dispersed in an upstream direction. Only thirty-six (<10%) of 389 individuals across all sites were identified as $F_0$ migrants using the "detection of first generation migrants" option in GENECLASS2 (Table 5).

## MtDNA sequences analysis

The edited alignment for the cyt *b* gene was 575 bp and included 121 variable positions. All sequences are lodged under GenBank accession numbers MG867590–MG867657. The neighbour-joining tree revealed two strongly supported clades (bootstrap 89% SEQ-S; 96% SEQ-N; Fig. 4). Phylogeographic structure was also clearly evident within clades. All individuals from four sites in the Brisbane River formed a distinct clade, and all three rivers sampled for the SEQ-N lineage formed shallow clades (i.e., Mary, Noosa and Mooloolah rivers; Fig. 4). Genetic distance was high between northern smelt lineages and the southern smelt sequences used as outgroups (uncorrected mean nucleotide distance 0.15–0.17). The mean nucleotide distance between the two northern lineages (SEQ-N and SEQ-S) was 0.04 (SE = 0.007).

## DISCUSSION

### Population structure and dispersal

Based on previous studies of Australian smelt (*Woods et al., 2010*; *Hughes et al., 2014*), we had hypothesized that *R. semoni* in south-east Queensland would exhibit limited genetic connectivity among river systems. Our findings of strong genetic differentiation among rivers support this hypothesis. For both groups (SEQ-N and SEQ-S), there were highly significant $F_{ST}$ values, which indicated that populations were not panmictic. The results of the significant genetic differentiation among rivers were also consistent with the results of population structure revealed from the STRUCTURE analysis, suggesting restricted gene flow and limited dispersal among populations in both groups. Limited dispersal

**Table 4  Estimates of contemporary migration rates (m) for each populations based on microsatellite data.** Migration rate estimates were based on unidirectional assignment of microsatellite genotypes in BAYESASS v1.3. Proportion of nonmigrant values represents the proportion of individuals assigned back to their sampling site of origin. Mean and 95% CI (confidence interval) values for uninformative data were 0.833 (0.675, 0.992). Putative source of migrants represents source population supplying migrants into the focal population where the estimated migration rate exceeds the upper 95% CI value for uninformative data (0.0124). Migration rates and source of migrants were not provided for cases where the 95% CI of estimated migration rate overlapped with the 95% CI value for uninformative data.

| Focal populations (Site code) | Proportion of nonmigrants | Putative source of migrants | Migration rate (m), mean (95% CI) |
|---|---|---|---|
| Tinana Creek (MRD) | 0.988 | – | – |
| Mary_upper (MRU) | 0.987 | – | – |
| Noosa_lower (NSD) | 0.990 | – | – |
| Mooloolah_lower (MLD) | 0.685 | Mooloolah_upper (MLU) | 0.240 (0.130, 0.317) |
| Mooloolah_upper (MLU) | 0.990 | – | – |
| Brisbane_lower (BRD) | 0.677 | Brisbane_upper (BRU) | 0.287 (0.232, 0.325) |
| Brisbane_upper (BRU) | 0.989 | – | – |
| Logan/Albert_lower (LAD) | 0.699 | Logan/Albert_upper (LAU) | 0.145 (0.033, 0.279) |
| Logan/Albert_upper (LAU) | 0.987 | – | – |
| Coomera_lower (CMD) | 0.681 | Coomera_upper (CMU) | 0.271 (0.198, 0.323) |
| Coomera_upper (CMU) | 0.990 | – | – |
| Nerang_lower (NRD) | 0.668 | Nerang_upper (NRU) | 0.148 (0.036, 0.281) |
| Nerang_upper (NRU) | 0.990 | – | – |
| Currumbin_lower (CRD) | 0.990 | – | – |
| Currumbin_upper (CRU) | 0.677 | Currumbin_lower (CRD) | 0.286 (0.235, 0.326) |

was supported by our first-generation migrant detection analysis in Geneclass2, which demonstrated that less than 10% of individuals in each population were immigrants.

The sample from Tinana Creek (MRD site), was differentiated from the rest of the populations in the SEQ-N group (Table 2). This might be the result of a barrier which separates Tinana Creek from the rest of the Mary River system despite their close proximity to one another. Tinana Creek runs into the Mary River not far from the mouth, with both drainages having tidal estuarine reaches in the lower sections. The differentiation of the Tinana Creek population from the main stem of the Mary River is also observed in a number of other freshwater species including Mary River Cod, *Maccullochella mariensis* (*Huey, Espinoza & Hughes, 2013*), Mary River Turtle, *Elusor macrurus* (*Schmidt et al., 2018*), freshwater crayfish *Cherax dispar* (*Bentley, Schmidt & Hughes, 2010*) and Australian lungfish *Neoceratodus fosteri* (*Hughes et al., 2015*).

Although populations in the SEQ-N group were slightly more highly structured than those in the SEQ-S group, fishes in both groups exhibited restricted gene flow. The high genetic structuring in SEQ-N populations might be the result of genetic divergence within the lineage that occurs over a very small geographical range. The lower levels of genetic structure of *R. semoni* populations in the SEQ-S group than the SEQ-N group suggests that more current or recent gene flow occurs within and among the catchments in SEQ-S group than for the SEQ-N group. This could be because the catchments in the SEQ-S group have

Islam et al. (2018), *PeerJ*, DOI 10.7717/peerj.4654

**Table 5  Results of the assessment for detecting first-generation migrants performed using GENECLASS2 showing the number of individual migrants ($P < 0.05$) detected per sampling location.** Results are based on the L_home/L_max statistic for microsatellite data. L_home/L_max is the ratio of the likelihood computed from the population where the individual was sampled (L_home) over the highest likelihood value among all population samples including the population where the individual was sampled. $F_0$ is the first generation migrant. See Table 1A for site codes and locations.

| Sample to | $F_0$ Migrants from | | | | | | | | | | | | | | |
|---|---|---|---|---|---|---|---|---|---|---|---|---|---|---|---|
| | MRD | MRU | NSD | MLD | MLU | BRD | BRU | LAD | LAU | CMD | CMU | NRD | NRU | CRD | CRU |
| MRD | | 0 | 0 | 0 | 0 | 0 | 0 | 0 | 0 | 0 | 0 | 0 | 0 | 0 | 0 |
| MRU | 0 | | 0 | 0 | 0 | 0 | 0 | 0 | 0 | 0 | 0 | 0 | 0 | 0 | 0 |
| NSD | 0 | 0 | | 0 | 0 | 0 | 0 | 0 | 0 | 0 | 0 | 0 | 0 | 0 | 0 |
| MLD | 0 | 0 | 0 | | 2 | 0 | 0 | 0 | 0 | 0 | 0 | 0 | 0 | 0 | 0 |
| MLU | 0 | 0 | 0 | 0 | | 0 | 0 | 0 | 0 | 0 | 0 | 0 | 0 | 0 | 0 |
| BRD | 0 | 0 | 0 | 0 | 0 | | 1 | 0 | 0 | 0 | 0 | 0 | 0 | 0 | 0 |
| BRU | 0 | 0 | 0 | 0 | 0 | 1 | | 0 | 0 | 0 | 0 | 0 | 0 | 0 | 0 |
| LAD | 0 | 0 | 0 | 0 | 0 | 0 | 0 | | 4 | 0 | 0 | 0 | 0 | 0 | 0 |
| LAU | 0 | 0 | 0 | 0 | 0 | 0 | 0 | 3 | | 0 | 0 | 0 | 0 | 0 | 0 |
| CMD | 0 | 0 | 0 | 0 | 0 | 0 | 0 | 0 | 0 | | 6 | 0 | 0 | 0 | 0 |
| CMU | 0 | 0 | 0 | 0 | 0 | 0 | 0 | 0 | 0 | 6 | | 0 | 0 | 0 | 0 |
| NRD | 0 | 0 | 0 | 0 | 0 | 0 | 0 | 0 | 0 | 0 | 0 | | 2 | 0 | 0 |
| NRU | 0 | 0 | 0 | 0 | 0 | 0 | 0 | 0 | 0 | 0 | 0 | 1 | | 0 | 0 |
| CRD | 0 | 0 | 0 | 0 | 0 | 0 | 0 | 0 | 0 | 0 | 0 | 0 | 0 | | 6 |
| CRU | 0 | 0 | 0 | 0 | 0 | 0 | 0 | 0 | 0 | 0 | 0 | 0 | 0 | 4 | |
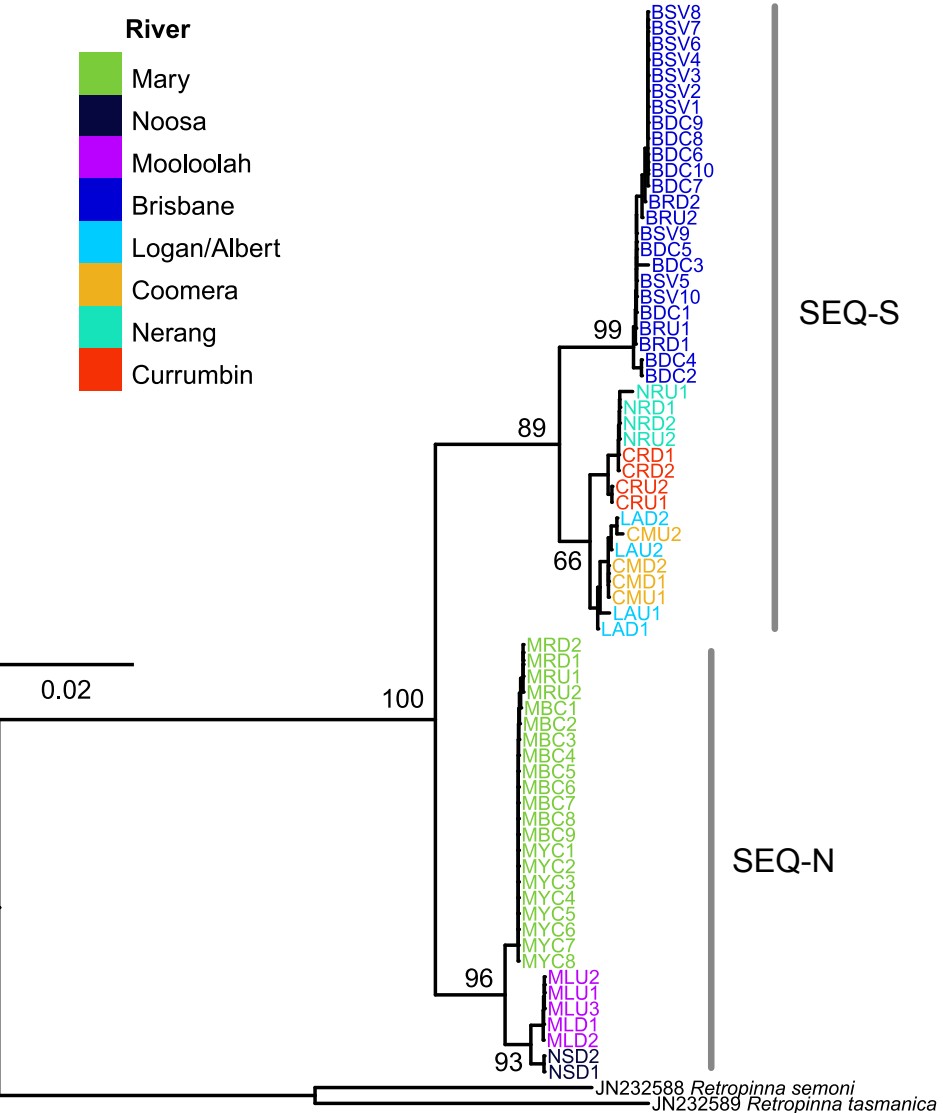

**Figure 4** **Neighbour-joining tree of the cyt *b* dataset for 68 Australian smelt samples from 19 sampling localities.** Individual sample codes coloured according to river. Node values are bootstrap support. See Table 1B for site codes and locations. Individual sample codes coloured according to river. Node values are bootstrap support. See Table 1B for site codes and locations.

been connected more recently and/or more often than those in the SEQ-N group. This could be because during extreme flood events, SEQ-S catchments might be connected by freshwater plumes flowing into Southern Moreton Bay. Another plausible reason might be that these cryptic species (SEQ-N and SEQ-S) have different microhabitat preferences, with SEQ-N being less tolerant of brackish conditions (*Hughes, 2015*).

An important model for stream dwelling species is isolation by distance (IBD). In this model, equilibrium between genetic drift and gene flow may be reached in species where the life time dispersal distance is less than the range. Here, a relationship between stream

distance and genetic differentiation should be evident (*Wright, 1943*). In this study, a strong IBD relationship was identified among the SEQ-S populations, but not among SEQ-N populations. This suggests that for SEQ-S populations, dispersal, when it occurs, is more likely between nearby catchments. Similar IBD relationships have been reported for other coastline restricted species (*Keenan, 1994*; *Jerry & Baverstock, 1998*; *Shaddick et al., 2011*; *Schmidt et al., 2014*). Lack of IBD for the SEQ-N group may be attributed to an insufficient number of population samples available for comparison and/or the greater degree of population isolation within this group relative to the SEQ-S group, consistent with the overall higher $F_{ST}$ estimates among SEQ-N populations. *Hughes et al. (2014)* observed similarly contrasting patterns of population genetic structure between cryptic species groups of southern Australian smelt. In that study, two informal species groups (MTV and SEC) with adjacent distributions along the western and eastern coast of southern Victoria had microsatellite-based $F_{ST}$ values of 0.19 and 0.07 respectively (*Hughes et al., 2014*). Using otolith microchemistry, *Hughes et al. (2014)* also showed that the more structured western group (MTV) had a greater proportion of nondiadromous populations relative to the weaker structured eastern group (SEC). The similar pattern of contrasting structure observed here between northern groups in the Australian smelt complex (SEQ-N and SEQ-S), is probably not due to differences in diadromous behaviour because preliminary evidence from otolith chemistry suggests all of these populations are nondiadromous (R Islam, 2017, unpublished data). Higher structuring of the SEQ-N group could possibly be due to genetic drift if these populations have been established for a longer period of time at the northern-most limit of Australian smelt distribution relative to the SEQ-S populations.

The complementary pattern of population structuring in both microsatellite and mtDNA data between the SEQ-N and SEQ-S groups could reflect long-term phylogeographic isolation or perhaps further evidence for cryptic species within Australian smelts as first highlighted by *Hammer et al. (2007)*. Mean cyt *b* divergence of 4% between SEQ-N and SEQ-S samples is close to the 3.6% divergence observed for the full mitochondrial molecule reported by *Schmidt, Islam & Hughes (2016)*, and is within the range of lineage divergence reported for *R. semoni* in southern Queensland (*Page & Hughes, 2014*). It should be noted that *Schmidt, Islam & Hughes (2016)* incorrectly attributed mitochondrial sequence data for the SEQ-N group to the CEQ group (i.e., Genbank accession KX421785 corresponds to SEQ-N and KX421784 corresponds to SEQ-S). The level of cyt *b* divergence between the SEQ groups and southern lineages of *R. semoni* is very large (15–17%) and adds to previous studies that have highlighted the likely existence of a cryptic species complex within the taxon currently referred to as *R. semoni* (*Hammer et al., 2007*; *Hughes et al., 2014*).

## Contemporary migration

The Bayesian assignment analysis detected contemporary movement of individuals only between proximate sites within rivers (Table 4). Contemporary dispersal was not observed between rivers. Although, most of the sites that we sampled were within 10–60 km stream distance of another sampled site, there was no contemporary dispersal among the majority of those rivers in either group. The present results coincide with the findings of some southern smelt populations, where contemporary movement was observed among

populations within a catchment (*Woods et al., 2010*). The species appears to occur in a wide range of freshwater habitats, many of which are isolated by long stretches of unfavourable habitat. Our data therefore suggest that if local extinctions occur in one or more of these streams within a reach of the river, then recolonization from elsewhere is unlikely to occur rapidly.

## CONCLUSION

Finally, the present study revealed high levels of population structuring within and between drainages, which suggested that contemporary movement among sites was rare and limited to sites within the same river. Little conservation attention has been given to the Australian smelt since it has long been considered a common species distributed widely across south-eastern Australia. The findings of the present study and other recent research (*Hammer et al., 2007*; *Crook, Macdonald & Raadik, 2008*; *Hughes et al., 2014*) suggest that southern smelts are a genetically complex and ecologically diverse taxonomic group. Therefore, proper conservation and management will require appropriate taxonomic treatment to align species with the clear genetic divisions now recognised across the range of Australian smelt.

Further, eight isolated management units (MUs) along the south-east Queensland drainages were detected in *R. semoni* from the microsatellite dataset (Fig. 2) demonstrating little to no gene flow between them. These management units align with individual coastal catchments, which suggest that other genetically distinct populations may exist in coastal rivers not sampled in this study. High levels of genetic divergence between the two lineages have important implications for the conservation of these endemic freshwater cryptic fish species. Therefore, findings of the present study on population structure of Australian smelt will help to formulate effective management and conservation plans for this cryptic species complex across their geographic range.

## ACKNOWLEDGEMENTS

We thank Leo Lee, Nathan McIntyre, Dale Bryan Brown, Juan Tao for help in collecting samples. Thanks to Kathryn Real and Jemma Somerville for assistance with genetic laboratory work.

### Funding

Funding for this research was provided by Griffith University to Md Rakeb-Ul Islam through International Postgraduate Research Award. The funders had no role in study design, data collection and analysis, decision to publish, or preparation of the manuscript.

### Grant Disclosures

The following grant information was disclosed by the authors:
Griffith University.

## Competing Interests

Jane M. Hughes is an Academic Editor of PeerJ.

## Author Contributions

- Md Rakeb-Ul Islam conceived and designed the experiments, performed the experiments, analyzed the data, prepared figures and/or tables, approved the final draft.
- Daniel J. Schmidt analyzed the data, prepared figures and/or tables, authored or reviewed drafts of the paper, approved the final draft.
- David A. Crook authored or reviewed drafts of the paper, approved the final draft.
- Jane M. Hughes conceived and designed the experiments, contributed reagents/materials/analysis tools, authored or reviewed drafts of the paper, approved the final draft.

## Animal Ethics

The following information was supplied relating to ethical approvals (i.e., approving body and any reference numbers):

All field and experimental protocols carried out in this study were approved by the Griffith University Animal Ethics Committee.

## Data Availability

Microsatellite raw data and GenBank accession numbers of Cyt b haplotypes are available in Supplemental Files.

## Supplemental Information

Supplemental information for this article can be found online at http://dx.doi.org/10.7717/peerj.4654#supplemental-information.

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
