# Peer review of "Patterns of genetic structuring at the northern limits of the Australian smelt (Retropinna semoni) cryptic species complex"

_PeerJ, doi:10.7717/peerj.4654_

## Round 0.1 · original submission · Major Revisions

Please address all the comments and suggestions by the two reviewers and resubmit

·

Basic reporting

no comment

Experimental design

no comment

Validity of the findings

no comment

Additional comments

Islam et al. address population genetic structure of a northern group of the Australian smelt. They found high levels of genetic differentiation both among and within rivers, suggesting limited connectivity among populations.
Although the scientific novelty of this study is limited, their findings would be informative to the local resource managers. They hired standard population genetic procedures to find clear genetic differentiations among populations, and their conclusion seems to be reasonable. Their writing is clear in general.

Specific comments

Ln. 38: Is the comma necessary?

Ln. 85-104: I don’t find a strong connection between the theme of the manuscript and the species diversity in Australia argued here.

Ln. 113: What about the faithfulness of the homing behavior in this species? Given that even a small amount of reproduction by straying fish can homogenize genetic compositions between populations, it is meaningful to inform readers how much is known about the stray rates of this and the related species.

Ln. 130: Italic for gene name? (throughout the manuscript)

Ln. 179: fifteen or 15? Be consistent throughout the manuscript.

Ln. 247: Use citations for acknowledging the model and the method, even though they are implemented in a software.

Ln 258: Share which populations/loci exhibited deviations from HWE with readers, so as the instances of null alleles and significant LDs.

Ln. 283: What does ‘initially’ mean here? The authors argue that the highest likelihood at K=8 in Ln. 289.

Ln. 323: Provide accession numbers before the acceptance of the manuscript.

Table 1: Inclusion of population samples with n=8 in population genetic structure analyses is misleading. Try with and without these population samples and argue if their conclusions are supported even without these samples. It may also be reasonable to pool upper-lower population samples for these two cases, given the low Fst values within each river.

Table 5: What if P<0.05 was employed for relaxing the detection criteria? Sharing the data is ideal, and an argument on the selection criteria in the manuscript is a must.

Fig. 5: Use different colours for Mary and Coomera. Light yellow characters for Logan/Albert are hard to read.

Reviewer 2 ·

Basic reporting

Some of the punctuation and grammar was sub-optimal, leaving the reader confused about the points being made. There is inconsistent formatting of large numbers, inappropriate CAPS in several instances and scientific names that are not italicized.

I have provided commentary in my review attachment on instances where it needs to be improved.

I found parts of the introduction and discussion to be redundant. The authors may feel they were providing context to their study. But I consider the context provided didn't really hit the mark of explaining what it was that their study was trying to achieve.

There are some basic errors in simple parameters such as the family name of the study species, latitude swapped with longitude in Table 1 and an error in the tally of mtDNA samples processed. If the authors erroneously reported these data, how can the reader trust the accuracy of the more complicated figures?

The Genbank accession numbers where not provided in the manuscript. Only place-filling XXXXXX-XXXXX was provided.

The captions to the tables and figures were generally not sufficiently detailed to describe what was presented.

The resolution of figure 3 needs to be improved.

Suggestions for improvement are provided in my attached review.

Experimental design

Semi-original research that confirmed pre-existing knowledge on population structure of the relevant populations.

The authors clearly stated their hypothesis and posed the question of population structure relevant to life history strategies in previously un-studied northern populations of their study species (which was new). But they didn't really explain why this information is necessary or important.

The analyses used were rigorous. However, the authors could have made greater attempts to achieve their target sample sizes per population (this is an abundant species that is easy to catch) and they could have scrutinised the outputs of their analyses more thoroughly given what looks like one or two aberrent individuals in one population which are likely to have substantially contributed towards the results reported.

The methods were described in sufficient detail to be replicated.

Validity of the findings

Sample sizes were quite small in a few populations.

One or two aberrent individuals in one population are likely to have substantially contributed towards the results observed. The data from this/these individuals should be scrutinised. If it is found to be of questionable origin, then this sample should be omitted from the data and the data re-analysed.

The conclusions based on the objectives and hypotheses could be emphasised a little more. At present, the conclusions focus on the definition of units of conservation and don't relate all that well to the theme of dispersal and life history strategies.

Additional comments

Line 26: Family is Retropinnidae, not Retropinninae.
Line 26: It can’t simply be defined as a facultative diadromous species? The next sentence suggests other proposed definitions of movement behaviour, of which facultative diadromy is only one.
Line 32: The hypothesis is fine, but seems at odds with the description of life history/movement behaviour suggested (facultative diadromous). It would be fine under the assumption that they are more inclined to potamodromy rather than diadromy. It is explained pretty well in the sentence starting on line 127. Could you rephrase the hypothesis in the context of the objectives?
Line 38: The sentence starting on this line should be shifted up to precede the previous sentence.
Line 40: Before making this claim, you need to demonstrate that there are no natural or artificial fish passage barriers between your within-river sampling sites.
Line 42: Delete ‘Overall,’
Line 42: Insert ‘Australian’ before ‘smelt’.
Line 42: Insert ‘comparatively’ before ‘higher’.
Line 45: Specify that the 8 MUs are within Queensland. There are certainly much more at a national scale.
Line 64: The sentence that starts on this line needs punctuation.
Line 65: Replace ‘reproduction’ with ‘reproducing’.
Lines 57-72: Is the 1st paragraph necessary? I don’t think it adds any value to the reader.
Line 82: You don’t need the word ‘genetically’ here.
Line 88: You don’t need the word ‘as’ here.
Lines 85-104: Is the 2nd paragraph necessary? I don’t think it adds any value to the reader.
Line 106: The genus should be called the Southern smelts given that Australian smelt is specific to R. semoni. Family is Retropinnidae, not Retropinninae. The sentence should read “Southern smelt (Retropinnidae: Retropinna) are abundant fishes distributed …..”
Line 109: Southern smelts are currently recognised …..
Line 126: Add comma after ‘individuals’.
Line 128: replace ‘of’ with ‘in’.
Lines 129-130: Sentence should start “Mitochondrial DNA sequence data combined with nuclear data .…”
Lines 131-133: These hypothesese are fine – albeit a little uninspiring. Why is this description of the hypotheses different from that in the abstract?
Line 139: Replace “each river except ….” with “each of eight river systems (except the Noosa River – downstream only).”
Line 140: Delete “Where possible,” and start sentence with We. It was always possible to catch more than 30 (as you aimed to do). You just didn’t manage to.
Line 147: Delete “All subsequent”. That can be inferred.
Line 148: PCR acronym used 1st here, but spelled out on line 151,
Line 152: Check the journals format for the o superscript in degrees celcius. You have used the number 0. They may want a lower case o.
Line 154: delete “amplified”. That can be inferred.
Lines 159-164: Why aren’t these samples shown in Table 1. Wouldn’t that be simpler. You could have a part a and part b for usat v mtDNA samples.
Line 164: (2 x 15) +10+10+9+8 = 67. Where did the extra sample come from?
Line 176: Replace ‘calculated’ with ‘undertaken’.
Line 180: ARLEQUIN v3.5.1.2 (Excoffier & Lischer, 2010) is cited in full here. Does it need to be cited in full in every subsequent usage (ie. line 194, 198, 225). Or just ARLEQUIN as on line 190?
Line 182: Is ARpriv equivalent to what is labelled PAR in table 1? If so, use a consistent symbology.
Line 183: Replace ‘estimated’ with calculated’.
Line 191: Replace ‘by’ with ‘through’.
Line 191: You have used a comma as a 1,000s place marker in this large number 10,000 (and others). But you have not in several other places (ie. line 196, 224, 236, 248 etc). Be consistent.
Line 201: Add “eight groups,” immediately after b).
Line 202: It is unclear what “according to the connectivity of streams to the upper river reaches” means.
Line 225: Arlequin is in lower case here. Should it be capitalised?
Line 231: Add comma after ‘generations’.
Line 238: End sentence at “… 2003).” Restart next sentence at “To estimate …”
Line 239: Was the analysis singular or should 3 repeat analyses be plural?
Line 239: Is “different random number of seeds” accurate wording? Do you use a number of seeds? Or several different seed numbers?
Line 243: Add “the” before “Paetkau”
Line 247: Lower case N in neighbour-joining.
Line 251: if JN232588 lacks locality information, why did you use it. You haven’t justified your speculation that it belongs to a southern lineage. Why not ask Chris Burridge?
Line 253: Shouldn’t the reference be to Hughes et al. 2014 rather than 2015?
Line 267 and 269: What is the difference between Mean number of alleles and allelic richness averaged across loci?
Line 269-270: Why was it necessary to standardize sample sizes across populations at 6 individuals. Please explain.
Line 283: Be careful with the word “initially”. Did STRUCTURE determine this? Later it appears that STRUCTURE revealed the highest likelihood of 8 clusters. Both sentences can’t be correct. It is very confusing.
Line 285-287: It is interesting that the STRUCTURE outputs are counter to the FST results. Based on FST you would have expected the CEQ pops to separate in STRUCTURE before the SEQ populations. Why did this happen? Can the authors speculate?
Line 286-287: The second half of this sentence needs to be reworded. It is unclear.
Line 298-300: The sentence spanning these lines is very confusing – switching between singular and plural. Pleas reword. And define what ‘they”are/is.
Line 300-301: Again – counter to what happened in STRUCTURE.
Line 301: Replace “in” with “within”
Line 315: Capitalise C in Creek and replace “was” with “were”.
Line 323: GenBank accession numbers are “XXXXXX-XXXXXXX”.
Line 325-327: The result is consistent with STRUCTURE result rather than FST.
Line 334-335. Delete “using mtDNA and microsatellites”. It can be inferred.
Line 343: Delete “ of R. semoni”. It can be inferred.
Line 347-348: Was Tinana Creek (MRD) always differentiated because of the influence of the one (or perhaps 2) distinctly different individual(s) within the sample. Could this individual represent a mis-labelling error? What happens to the results if this individual is omitted from all the analyses? This is a serious issue that needs to be checked. Why didn't the BAYESASS and GENECLASS 2 analyses pull out these individuals as migrants when STRUCTURE clearly assigns them as such?
Line 349-350. Why is the reference to Hughes et al. (2015) relevant here?
Line 353: Lower case m in mariensis.
Line 355: Cherax disper = Cherax dispar. Is lung fish = lungfish?
Line 358-374: I found this paragraph lacked focus. It is hard to determine what point you are actually trying to get across. It starts OK = CEQ more highly structured than SEQ. But none of the subsequent sentences actually address issues relevant to why CEQ and SEQ differed.
Line 358: So why did STRUCTURE pull out the Brisbane River from within the SEQ populations before splitting any of the CEQ populations?
Line 366: Please define ’eustasy’.
Line 376: An alternative model of what? Please define.
Line 407: Replace “relative to lineages of R. semoni from the south of this species range” with “relative to more southern lineages of R. semoni”.
Line 415: Is the 10-60 km figure linear distance (crow flies) or river distance or between estuary mouths?
Line 417: “this species appears to occur in pools” is inaccurate speculation. They occur in all freshwater habitats. In fact, they are often most abundant in run and riffle habitats relative to pool habitats.
Lines 418-421. Evidence of migration within rivers does not constitute evidence of potamodromous migration. Dispersal within a river system can occur without it being a migration – which requires a regular population-level movement for a specific purpose.
Line 422: Reword – “where large number of Australian smelt was found…”
Line 421: Replace ”suggesting” with “suggests”.
Line 424-427: This sentence needs to be reworded.
Line 430-447: While these statements are appropriate, they are too off-topic to constitute a conclusions section in the paper.
Line 433: Replace “Australian” with “Southern”.
Line 435: Delete “names”
Line 438: Delete “recognized that are geographically”.
Line 439: Delete “ these lineages”.
Line 443: Reword “where conservation strategy should be specified accurately”
Line 444: You haven’t established that anyone is actually proposing any translocations. Add context or delete.
Line 449: Replace “Alternatively” with “Further”. ESUs and Mus are not mutually exclusive. And specify that there 8 MUs in Queensland. There will be many more nationally.
Line 452: catchment should be plural.
Line 464: Mogurnda should be italicised.
Line 479: Volume should be bold font.
Line 509: Check that Crook used Retropinninae rather than Retropinnidae.
Line 524: Indentifying mis-spelled.
Line 549: Lower case m in mariensis.
Line 597: Add a space between genus and name of Craterocephalus stercusmuscarum.
Line 601: Italicise Hypseleotris compressa.
Line 620: Volume number should be bold font.
Line 634: Volume number should be bold font.
Line 669: Italicise Hypseleotris.
Line 674: Volume number should be bold font.
Figure and Table captions are all generally insufficiently detailed and should be expanded upon.
Figure 1: This is not a map of Queensland. It is a map of south east Queensland spanning the Gold Coast to southern Fraser Coast. Major rivers and towns need to be added.
Figure 3: The resolution of the figure is poor.
Table 1: Lower case s in sample. Neither PAR or Population specific FST are mentioned in the caption. Perhaps specify that these statistics are for microsatellite loci. The latitude and longitude values do not coincide with the column headings. Switch them.
Table 2: Capitalise B in Bonferroni.
Table 3: Define FST, FCT and FSC in the caption.
Table 4: One what basis are the most relevant results most relevant?
Table 5: migrant = migrants. End sentence after “…. location.”. Restart next sentence at “Results ….”.

---

## Round 0.2 · accepted · Accept

Thank you for addressing all the required corrections and necessary information as suggested by the reviewers'. hope it will publish as soon as possible. Congratulations! and Hope you will also consider PeerJ for your next manuscript!

·

Basic reporting

I think it is ready for publication now, with one minor correction in Ln. 196 (‘micosatellite’ -> ‘microsatellite’).

Experimental design

ok

Validity of the findings

acceptable

Additional comments

I think it is ready for publication now, with one minor correction in Ln. 196 (‘micosatellite’ -> ‘microsatellite’).